# Infliximab Induced a Dissociated Response of Severe Periodontal Biomarkers in Rheumatoid Arthritis Patients

**DOI:** 10.3390/jcm8050751

**Published:** 2019-05-26

**Authors:** Mélanie Rinaudo-Gaujous, Vincent Blasco-Baque, Pierre Miossec, Philippe Gaudin, Pierre Farge, Xavier Roblin, Thierry Thomas, Stephane Paul, Hubert Marotte

**Affiliations:** 1GIMAP EA3064, Laboratory of Immunology and Immunomonitoring, CIC CIE3 Inserm Vaccinology, Hôpital Nord, CHU Saint-Etienne, 42270 Saint-Priest-en-Jarez, France; melanie.rinaudogaujous@chpg.mc (M.R.-G.); stephane.paul@chu-st-etienne.fr (S.P.); 2Institute of Cardiovascular and Metabolic Diseases, CHU Rangueil, 31400 Toulouse, France; vincent.blasco@inserm.fr; 3Clinical Immunology Unit, Departments of Immunology and Rheumatology, Hôpital Edouard Herriot, CHU Lyon, 69003 Lyon, France; pierre.miossec@univ-lyon1.fr; 4Department of Rheumatology, CHU Grenoble, 38130 Échirolles, France; pgaudin@chu-grenoble.fr; 5Faculty of Odontology, University Lyon I., 69622 Villeurbanne, France; pierre.farge@univ-lyon1.fr; 6Department of Gastroenterology, Hôpital Nord, CHU Saint-Etienne, 42270 Saint-Priest-en-Jarez, France; xavier.roblin@chu-st-etienne.fr; 7Department of Rheumatology, Hôpital Nord, CHU Saint-Etienne, 42055 Saint-Etienne, France; thierry.thomas@chu-st-etienne.fr; 8INSERM U1059, Université de Lyon–Université Jean Monnet, 42023 Saint-Etienne, France

**Keywords:** Rheumatoid arthritis, *Porphyromonas gingivalis*, periodontal disease, matrix metalloproteinase 3, infliximab

## Abstract

Objective: Rheumatoid arthritis and periodontal disease are associated together, but the effect of therapy provided for one disease to the second one remained under-investigated. This study investigated effect of infliximab therapy used to treat rheumatoid arthritis (RA) on various biomarkers of periodontal disease (PD) severity including serologies of *Porphyromonas gingivalis* and *Prevotella intermedia* and matrix metalloproteinase 3. Methods: Seventy nine RA patients were enrolled at the time to start infliximab therapy and the 28 joint disease activity score (DAS28), anti-cyclic citrullinated petides 2nd generation (anti-CCP2), anti-*P. gingivalis* antibody, and Matrix metalloproteinase 3 (MMP-3) were monitored before and at 6 months of infliximab therapy. Joint damage and severe periodontal disease were assessed at baseline. Anti-CCP2, anti-*P. gingivalis* antibody, and MMP-3 were determined by enzyme-linked immunosorbent assay (ELISA). Results: At baseline, anti-CCP2 titers were associated with anti-*P. gingivalis* lipopolysaccharide (LPS)-specific antibodies titers (*p* < 0.05). Anti-*P. gingivalis* antibodies were not significantly correlated with clinical, biological, or destruction parameters of RA disease. At 6 months of infliximab therapy, MMP-3 level decreased (from 119 ± 103 ng/mL to 62.44 ± 52 ng/mL; *p* < 0.0001), whereas *P. gingivalis* antibody levels remained at the same level. DAS28 and inflammation markers C-reactive protein (CRP) and Erythrocyte sedimentation rate (ESR) also decreased significantly during infliximab therapy (*p* < 0.05) as anti-CCP2 levels (*p* < 0.001). Only high MMP-3 level at baseline was associated with infliximab efficacy (*p* < 0.01). Conclusion: MMP-3 level can be a useful marker of the efficacy of infliximab in RA patients. The treatment did not affect anti-*P. gingivalis* antibodies.

## 1. Introduction

Rheumatoid arthritis (RA) is the most frequently chronic inflammatory joint disease characterized by synovial hypertrophy and inflammation with joint and subchondral bone destruction, which correlates with disability and loss of function [1]. Epidemiological studies suggest an association between RA and periodontal disease (PD) and confirmed by a recent meta-analysis [2]. Both diseases (RA and PD) share important similarities in their pathogenesis involving similar genetic background [3] or production of large amount of proinflammatory cytokines such as tumor necrosis factor alpha (TNF) [4]. In, P.D., inflammation is initiated and perpetuated by a subset of bacteria, including *Porphyromonas gingivalis* (*P. gingivalis*) and *Prevotella intermedia* (*P. intermedia*), two specific gram-negative anaerobic bacteria, which colonize the gingival sulcus and proliferate in the gingival plaque. The resulting chronic inflammatory response by the host induces destruction of the supporting structures of the teeth defining severe PD. *P. gingivalis* presence seems to be specific of severe PD. This was reinforced by recent data from an experimental rat model confirming the specific involvement of *P. gingivalis* in arthritis onset [5]. Furthermore, bacterial colonization was also described in the gut of RA patients [6]. 

Anti-citrullinated protein antibodies (ACPA) are the highest specific biomarker for RA diagnosis or prognosis and are now included in the new RA criteria [1]. Endogenous or exogenous peptidyl-arginine deiminases (PADs) induced citrullinated proteins by conversion of peptidyl-arginine to peptidyl-citrulline. This is part of many physiological processes [7]. However, smoking or *P. gingivalis* infection could induce excess of citrullination in some conditions [8]. Since ACPA occurred some years before RA clinical onset [9], *P. gingivalis* infection could precede RA onset and be a key player for initiation and maintenance of the autoimmune inflammatory responses in RA [10]. *P. gingivalis* is the unique known pathogen to have a specific enzyme PAD (PPAD) [11], which induce citrullination of proteins [12] and could provide a rupture of tolerance with ACPA induction. 

Indirect presence of *P. gingivalis* by serology demonstrated that high concentrations of anti-*P. gingivalis* antibody in established [12] or early RA patients [13]. This indirect biomarker of *P. gingivalis* correlated with the gingival bacteria load assessed by polymerase chain reaction [13,14]. PD is related to many other anaerobic periodontal pathogens including *P. intermedia*, which was also detected in both the serum and synovial fluids of RA patients [15]. 

Matrix metalloproteinase 3 (MMP-3) is one of the major MMPs expressed in rheumatoid synovial tissue [16]. MMP-3 is mainly involved in bone and cartilage degradation in RA or bone destruction in PD [17]. Thus, MMP-3 is already considered as a biomarker for RA and PD destruction [18]. In, P.D., a MMP-3 polymorphism was described as associated to PD [19]. Furthermore, strategy based on MMP-3 monitoring improved clinical response and reduced joint destruction in RA [20]. In both PD and, R.A., production of proinflammatory cytokines, as TNF., is increased and specific blocking of TNF improves two-third of RA patients [21]. Only few studies have reported predictive factors of response to infliximab in RA [21], but no relevant clinical or biological markers can be used in the daily practice. In only one previous study, PD was related to be a predictive factor for a non-response to TNF blocker therapy in RA patients [22]. Persistence of *P. gingivalis* in gingival tissue could participate to maintain local and systemic inflammation in relation with treatment resistance [23]. Only few studies explored therapeutic effect for PD on RA [24] and *vice versa* [25].

Since both diseases are associated at the susceptibility and severity level [24], therapy from one disease should be efficient to the second one. This concept was recently reinforce by the first demonstration of PD severity on RA activity [26]. We already reviewed previously [24] impact of some biologic disease modifying anti-rheumatic drugs (bDMARDs) on PD. Infliximab treatment worsened the gingival inflammation, but decreased the gingival destruction of bone [25]. A the opposite, rituximab [27] or tocilizumab [28] decreased gingival inflammation or gingival bone destruction related to the PD. Accordingly, in case of severe PD B-cell blocker or IL-6 receptor blockers could be considered preferentially compared to TNF blocker. At the opposite, some non-surgical PD therapy reported decreased of anti-*P. gingivalis* antibodies without effect on ACPA level [29]. 

Thus, our aim in this study was to correlate marker of PD severity (MMP-3, anti-*P. gingivalis* and anti-*P. intermedia* antibodies) and to assess effect of infliximab therapy on PD severe biomarkers in RA patients. In addition, the usefulness of these biomarkers was assessed for prediction of clinical response to infliximab therapy.

## 2. Patients and methods

### 2.1. Patients and Controls

Seventy nine RA patients treated with methotrexate with active disease and starting infliximab therapy were included consecutively. Following clinical parameters were recorded: Age, sex, disease duration, patient’s global assessment of disease activity, 28 tender and swollen joint counts, and the 28 joint disease activity score (DAS28). Joint damage and severe PD were defined by a right Larsen wrist score ≥ 2 and Hugoson and Jordan criteria, respectively as previously used [3]. Wrist X-rays were examined by the same reader (HM) as panoramic X-rays (PF). Clinical response to infliximab was defined by a decrease of DAS28 > 1.2. Blood samples were collected before and at 6 months of infliximab therapy to assess anti-cyclic citrullinated peptide second generation (CCP2), rheumatoid factor (RF), MMP-3, and antibodies against *P. gingivalis* and *P. intermedia*. Sera from two control groups were used in this study. We enrolled 27 healthy blood donors as control healthy volunteers and 28 patients with inflammatory bowel disease (IBD) and 35 patients with systemic lupus erythematosus (SLE), as inflammatory disorders controls. Local clinical ethics committee approved the protocol and all patients gave their written informed consent.

### 2.2. Methods

Determination of anti-*P. gingivalis* and anti-*Escherichia coli* LPS-specific antibodies by enzyme-linked immunosorbent assay (ELISA). To optimise our evaluation anti-*P. gingivalis* antibody assessed by ELISA., we performed two standardised ways for coating: Whole extract or lipopolysaccharide (LPS) components. LPS from *P. gingivalis* (InvivoGen, Toulouse, France) was coated on 96-well plates (Nunc, Dominique Dutscher, Brumath, France) at 10 µg/mL (diluted in carbonate buffer, pH 9.6) and incubated overnight at 4 °C. We used LPS from *Escherichia coli* (*E. coli*, InvivoGen, Toulouse, France) as control with the same dilution. Wells were washed three times with phosphate buffered saline (PBS)-Tween (0.005%). Plasma were diluted to 1:600 in PBS containing 1% of bovine serum albumin (BSA) and incubated (in duplicate) for 2 h at room temperature. Plates were washed as described above and incubated with peroxidase-conjugated goat anti-human IgG H + L (Jackson ImmunoResearch, West Grove, PA, USA) (diluted 1:50 000 in PBS) for 2 h at room temperature. After a final wash, detection was made by tetramethylbenzidine substrate (R&D Systems, Minneapolis, MN., USA). The reaction was stopped by the addition of H_2_SO_4_ (1M) solution and absorbances were measured at 450 nm. 

#### 2.2.1. Determination of Anti-P. gingivalis and Anti-P. intermedia Whole Extract Antibodies by ELISA

*P. gingivalis* strain ATCC 33277 and *P. intermedia* CIP 103607 were grown on sterile non-selective agar containing defibrinated sheep blood, supplemented with 0.0002% menadione sodium bisulfite and 0.4% hemin chloride. Cultures were then placed in an anaerobic chamber for 7 days at 37 °C. Then, colonies were recovered in a sterile water solution and centrifuged for 20 min at 5000× *g*. Pellets were diluted in sterile PBS at 50 mg/mL and were frozen at −20 °C until use. *P. gingivalis* or *P. intermedia* solution was then washed twice with carbonate buffer (pH 9.6) and diluted in the same buffer to obtain a solution of 1 McFarland (DensiCHEK plus, Biomérieux, Craponne, France). The solution was heated to 60 °C for 45 min, filtered (0.22 µm), diluted 1:10, coated on a 96-well plate and then incubated overnight at 4 °C. Plasma were diluted to 1:900 in PBS containing 1% BSA. Following steps are identical to those described for the LPS-specific ELISA. Cut-off values for seropositivity to *P. gingivalis* (LPS and whole extract) and *P. intermedia* were determined by concentrations higher than the 95th percentile in 27 healthy blood donors. 

A calibration curve was systematically done for each plate with dilutions of a pool of positive plasma diluted six times from 1:100 to 1:16200 to correct for plate-to-plate variation. Two plasma controls (high and low positives) were included on all plates. All intra assay coefficients of variation (CV) were below 6.5%. Inter assay CV for the high positive control were 13%, 8%, and 28% for *P. gingivalis* whole extract, LPS assay and *P. intermedia* assay, respectively. Results are expressed in Arbitrary Units (AU) defined by the pool dilution (10 AU = 1:16200 to 2430 AU = 1:100). 

Furthermore, determination of citrullinated proteins in *P. gingivalis* whole extract or LPS was assessed by using the specific anti-CCP2 detection antibody (Phadia, Thermo Fisher Scientific, Uppsala, Sweden).

#### 2.2.2. Assessments of ACPA, R.F., and MMP-3

MMP-3 blood concentrations were determined by a commercial ELISA method (AESKU.diagnostics, Wendelsheim, Germany). ACPA was assessed by anti-CCP2, and RF (IgA and IgM) were measured by ELIA method on ImmunoCap 250 (Phadia, Thermo Fisher Scientific, Uppsala, Sweden). Anti-CCP2 was considered to be positive at a cut off value of 10 U/mL., RF IgA at 14 IU/mL and RF IgM at 3.5 IU/mL, as recommended by the manufacturer. 

### 2.3. Statistical Analysis

Since data were not normally distributed, they were expressed as median and interquartile range 25–75% (IQR 25–75) or number (%). Correlations were performed by Spearman tests. Comparisons between controls and RA patients were performed by Mann Whitney test. Comparisons between baseline and 6 months of infliximab treatment were performed by Wilcoxon test. Statistics were done with the software GraphPad Prism (version 5.0). *p* values less than 0.05 were considered as statistically significant.

## 3. Results

### 3.1. RA Population 

Our RA population had the main characteristics of RA patients treated with TNF inhibitors as reported in the Table 1. In this cohort, severe PD was present in 51 RA patients and severe PD was associated with joint damage (*χ^2^* test = 4.4; *p* = 0.0276). 

### 3.2. Immunity Against Oral Pathogens is Related to Established RA

Anti-*P. gingivalis* whole extract antibodies were more frequently positive in established RA patients (97.5%) than in healthy blood donors (5%) with higher concentrations of anti-*P. gingivalis* antibody in established RA patients (238 (148–377) AU) than in healthy blood donors (43 (24–79) AU; Figure 1A; *p* < 0.001). Similar results were observed for anti-*P. gingivalis* LPS specific antibody (data not shown). Analogous pattern was also observed for anti-*P. intermedia* whole extract antibodies with more positive in established RA patients (84.8%) than in healthy blood donors (17.8%; Figure 1A; *p* < 0.001). Anti-*P. intermedia* antibody concentrations were also higher in established RA patients (390 (131–1558) AU) than in healthy blood donors (96 (67–179) AU; Figure 1A; *p* < 0.001). Thus immunity against oral pathogens was higher in established RA patients than in healthy blood donors reinforcing association between RA and PD. Furthermore, the two ways for anti-*P. gingivalis* antibody determination strongly correlated together (Figure 1B; *p* < 0.0001) and also correlated with anti-*P. intermedia* antibody concentrations (Figure 1C; *p* < 0.0001). Since we observed a correlation between immunity against these two bacteria from the oral cavity, we investigated the specificity of our assay. For this purpose, we tested the same plasma for anti-LPS fraction of *E. coli*, another commensal bacterium of the gut. Only one patient has serum with anti-*E. coli* antibody without anti-*P. gingivalis* antibody. At the opposite, many patients had anti-*P. gingivalis* antibody without anti-*E. coli* antibody, demonstrating the absence of crossreaction between these two antibodies (Figure 1D). As *P. gingivalis* whole extract may contain citrullinated proteins, we then investigated a putative crossreactivity between anti-CCP2 and anti-*P. gingivalis* antibodies. Presence of citrullinated proteins was not observed in the whole *P. gingivalis* extract by using the monoclonal antibody to detect anti-CCP2 (data not shown). 

### 3.3. Anti-P. gingivalis and anti-P. Intermedia Antibodies and RA Specificity

To assess the specificity of these antibodies to, R.A., we then assessed them in IBD and SLE., two other inflammatory auto-immune diseases. Anti-*P. gingivalis* and anti-*P. intermedia* antibody concentrations were lower in IBD and SLE patients and established RA patients (Figure 1E,F; *p* < 0.001 for *P. gingivalis* and *P. intermedia* in RA and SLE vs. IBD ).Thus high antibody concentrations against oral pathogens were specific for RA or SLE diseases involving joint. 

### 3.4. Differential Implication of P. Gingivalis and P. Intermedia Antibodies in Immune Response in RA Patients

Since high antibody concentrations against *P. gingivalis* and *P. intermedia* were observed in RA patients, association of these antibodies with clinical and biological RA parameters was investigated. RF IgA concentrations were correlated with DAS28 at baseline (Figure 2A; *p* < 0.01). Furthermore, anti-CCP2 concentrations correlated with both anti-*P gingivalis* (whole extract and LPS) antibody concentrations (Figure 2B,C; *p* < 0.001 and *p* < 0.01; respectively), without correlation with IgM RF concentrations. Interestingly, anti-*P intermedia* antibody concentrations correlated with IgM RF concentrations (Figure 2D; *p* < 0.05), but not with anti-CCP2 concentrations. Since only anti-*P gingivalis* concentrations correlated with anti-CCP2 concentrations, these data reinforced the association between *P. gingivalis* and citrullination in RA.

### 3.5. Clinical Response to Infliximab Therapy and Joint or Periodontal Destructions

As expected, DAS28 and inflammation markers (CRP and ESR) strongly decreased at 6 months of infliximab treatment (Table 1). DAS28 improvement was similar in RA patients with or without severe PD (1.9 (0.7–3.0) vs. 1.7 (0.5–2.5); not significant) and according to the joint damage status. Interestingly, anti-CCP2 concentrations decreased significantly during infliximab therapy (Figure 3A; *p* < 0.001) as IgM RF concentrations (Figure 3B; *p* < 0.001). As expected, all RA parameters decreased during infliximab therapy.

### 3.6. Dissociated Effect of Infliximab Therapy on PD Biological Markers

Anti-*P. gingivalis* whole extract antibody concentrations slightly increased at 6 months of infliximab therapy (Figure 3C; *p* < 0.05) with the same trend for anti-*P. intermedia* antibody concentrations (Figure 3D; *p* = 0.052). During infliximab therapy, MMP-3 concentrations strongly decreased from 90 (14–259) ng/mL to 45 (10–160) ng/mL (Figure 3E; *p* < 0.0001). Thus, infliximab therapy induced a strong reduction of MMP-3, but slightly increased concentrations of antibodies against *P. gingivalis* and *P. intermedia*.

### 3.7. Biological Markers and Prediction of Clinical Response

Among biomarkers assessed in this study, baseline MMP-3 concentrations were higher in patients with a good response to infliximab (111 (49–187) ng/mL) compared to non-responders (57 (29–110) ng/mL; *p* < 0.05; Figure 4A). However, neither baseline anti-CCP2 concentrations (Figure 4B; NS) nor anti-*P. gingivalis* and anti-*P. intermedia* antibody concentrations (data not shown) were not associated with clinical response to infliximab. Furthermore, reduction of MMP-3 during infliximab therapy was higher in responders than in non-responders patients (Figure 4C; *p* < 0.05). Only a trend was observed for higher reduction of anti-CCP2 in responders compared to non-responders to infliximab (Figure 4D; *p* = 0.09).

## 4. Discussion

We confirmed a higher prevalence of PD in established RA patients (64.5%) than in the general population (30–40%) [30], as previously reported in early RA [31]. PD is related to many anaerobic periodontal pathogens including *P. intermedia*, *P. melaninogenica*, or *T. forsythia*. Antibodies against these periodontal pathogens were also more frequently detected in the serum from RA patients than controls [32]. We first confirmed higher concentrations of antibodies against *P. gingivalis* and *P. intermedia* in established RA than in healthy blood donors [32] or in two other inflammatory diseases (SLE and IBD). This is in link with a recent review considering high *P. gingivalis* antibody as a biomarker of RA [33]. Serology determination is the easiest way to assess presence of *P. gingivalis* and indirectly presence of severe PD. Gingival presence of *P. gingivalis* detected by polymerase chain reaction is strongly correlated with concentrations of anti-*P. gingivalis* antibody [13,14]. Previous studies have explored antibodies against *P. gingivalis* in RA patients by using either LPS or whole extract [12]. Here, we observed a correlation between these two assessments (LPS and bacterial extract) of antibodies against *P. gingivalis*, suggesting that LPS should be the easiest way to assess anti-*P. gingivalis*. Furthermore, antibody concentrations against *P. gingivalis* and *P. intermedia* correlated together, which was not surprising since both are involved in PD pathogenesis [34]. Despite the quite high r value for the correlation between antibodies against *P. gingivalis* LPS and against *E. coli*, we still believe that each antibody is specific due to absence of high level of antibodies for both pathogens. Our assay to detect antibodies against *P. gingivalis* LPS or *E. coli* LPS without crossreaction. This was not surprising since LPS from *P. gingivalis* has a specific structure compared to other bacteria with the ability to activate TLR2 and TLR4 [35], whereas LPS from *E. coli* activates mainly TLR4 [35].

Anti-*P. gingivalis* antibody concentrations correlated with anti-CCP2 concentrations without correlation with RF concentrations [15]. We did not expected crossreacticvity between *P. gingivalis* and anti-CCP2 due to LPS structure containing only sugars and lipids and absence of detection of citrullinated proteins in *P. gingivalis* extracts by using the monoclonal detection antibody from anti-CCP2 assay. Contrary to anti-*P. gingivalis* antibodies, anti-*P. intermedia* antibodies were not correlated with anti-CCP2, but with RF IgM concentrations, as previously described in a human [36] or in a rat model [5]. This suggested different mechanisms of action between these two bacteria in immune response induction. Assessment of several oral bacteria is required to have a discriminate role for each pathogen. For instance, recent data suggested high anti-*P. intermedia* antibodies in clinical remission RA patients with persistent PD US activity [23]. On the other hand, *Aggregatibacter actinomycetemcomitans* (*A. actinomycetemcomitans*), a periodontal pathogen associated with PD severity, has been suspected to be the culprit of the association of RA and PD [37] without confirmation since. Taken all together, our data support the specific implication of *P. gingivalis* compared to *P. intermedia* in RA pathogenesis with induction of anti-CCP2 in response to gingival citrullinated proteins, as suggested by a recent rodent model [5].

MMP-3 is a biomarker of both destruction in RA [38] and in PD [39]. So, we confirmed the reduction of inflammatory markers (ESR., CRP) and MMP-3 over infliximab treatment, as previously described with infliximab [40] or other TNF blocker [41]. 

Despite the growing interest for *P. gingivalis* in, R.A., this was the first observation that anti-*P. gingivalis* and *P. intermedia* antibodies were increasing over time during infliximab therapy. Only a recent study observed a stability of anti-*P. gingivalis* antibody concentrations in 50 early RA patients mainly treated with methotrexate [42]. As previously reported with methotrexate therapy [42], no patient developed seroconversion for *P. gingivalis* or *P. intermedia* with combination of infliximab and methotrexate. Despite the growing interest for *P. gingivalis* pre-existing immunity in RA patients, this was the first observation that serum anti-*P. gingivalis* antibodies were increasing over time during infliximab therapy or bDAMRDS. It is not surprinsing that bDMARD promotes infections including gingival infection. Our results are in line with previous report with infliximab showing increasing of gingival inflammation during infliximab therapy [25]. However, this increasing of gingival inflammation was not observed with rituximab or tocilizumab, two other bDMARDs targeting B cell and IL-6, respectively [27,28]. On the contrary, some studies investigated the effect of periodontal therapy on RA activity. A recent metaanalysis suggested a small effect of periodontal therapy (full-mouth scaling and root planning) on RA disease activity [43]. Reduction of ACPA during infliximab therapy was also already described [44]. Despite increased of anti-*P. gingivalis* antibody concentrations during infliximab therapy, a trend for a reduction of anti-CCP2 during infliximab therapy was observed in our study. This kind of dissociation response was also observed after non-surgical therapy for periodontal disease with a decreased of anti-*P. gingivalis* levels without effect on anti-CCP2 levels [29]. This dissociated response to infliximab of anti-*P. gingivalis* and anti-CCP2 concentrations reinforced role of *P. gingivalis* only in anti-CCP2 induction. 

Prediction of TNF response remains a huge challenge and to date no biomarkers can be used in daily practice [21]. Here, we observed that high baseline MMP-3 concentrations were associated with a good clinical response. This emphasizes the role of MMP-3 for RA management strategy [20]. However, baseline MMP-3 concentrations were similar in 47 good responders vs. 29 non-responders according to response assessed earlier than in our study at 14 weeks [45]. Further studies are needed before to consider MMP-3 as a validate biomarker to predict clinical response in the daily practice [21]. 

Our study also has some weakness. PD assessment was only performed at baseline on panoramic X-rays without clinical assessment. Clinical PD assessment was not planned in this study. However, clinical severe PD was already extremely reported [2] with a dissociated effect of infliximab therapy. In fact, infliximab treatment worsens the gingival inflammation, but decreases the gingival destruction of bone [25]. Panoramic X-rays was not repeated due to low variation of bone loss. 

## 5. Conclusions

Concentrations of anti-*P. gingivalis* antibody, a biomarker of PD severity correlated with anti-CCP2 concentrations in one hand and anti-*P. intermedia* antibody concentrations in another hand. Besides, anti-*P. intermedia* antibody concentrations correlated only with RF concentrations suggesting different immunologic response induced by both oral bacteria. Whereas MMP-3 is strongly decreased by infliximab therapy, serology against oral pathogen slightly increased. Furthermore, MMP-3, another biomarker of PD and RA severity, is a predictive biomarker of response to infliximab therapy. Our data reinforce interest of PD in RA pathogenesis and RA therapeutic management. They also confirm the possible involvement of *P. gingivalis* in RA physiopathology due to the correlation with ACPA.

## Figures and Tables

**Figure 1 jcm-08-00751-f001:**
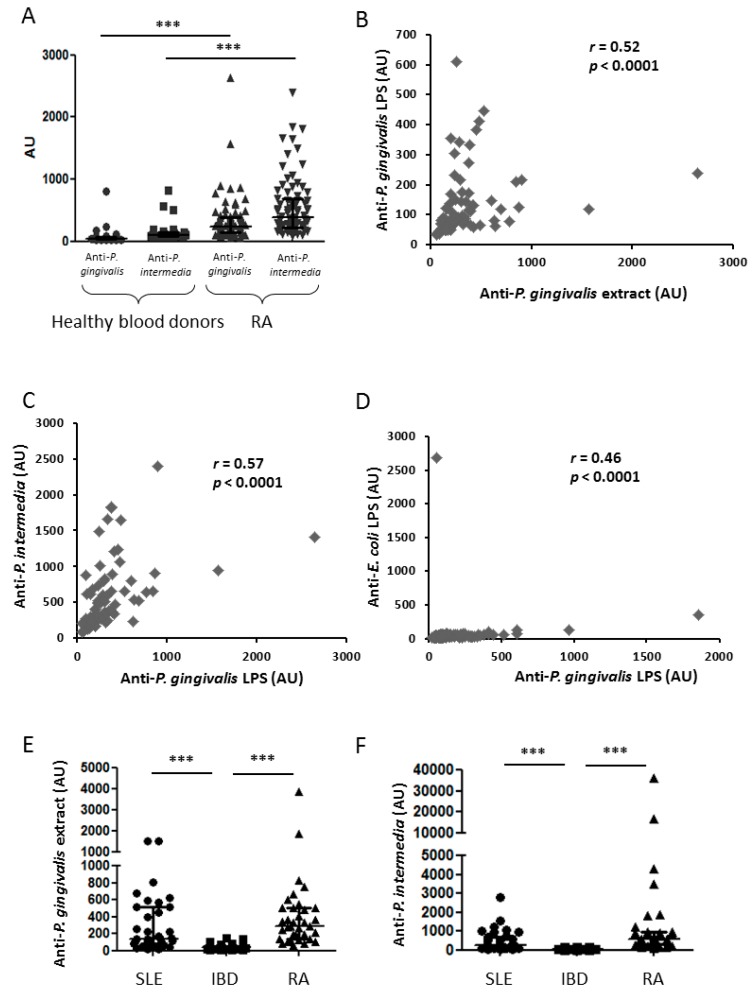
Evaluation of the measure of anti-*P. gingivalis* whole extract and LPS specific antibodies by ELISA. Anti-*P. gingivalis* and anti-*P. intermedia* (**A**) whole extract antibodies were measured in healthy blood donors and RA patients. Correlations between both anti-*P. gingivalis* LPS and whole extract (two ways to assess the same germ; (**B**); anti-*P. intermedia* and anti-*P. gingivalis* whole extract (assessment of two oral germs; (**C**) in RA patients. No correlation between anti-*P. gingivalis* and anti-*E. coli* LPS antibodies in RA patients (assessment of one oral germ and on commensal gut germ; (**D**). Anti-*P. gingivalis* (**E**) and anti-*P. intermedia* (**F**) whole extract antibodies were measured in SLE., IBD., and RA patients. From the bottom up, the bars indicate the interquartile range and the median. AU: arbitrary units; *r*: Spearman correlation coefficient; LPS: lipopolysaccharide; RA: rheumatoid arthritis; SLE: systemic lupus erythematosus; IBD: inflammatory bowel disease; ***: *p* < 0.001.

**Figure 2 jcm-08-00751-f002:**
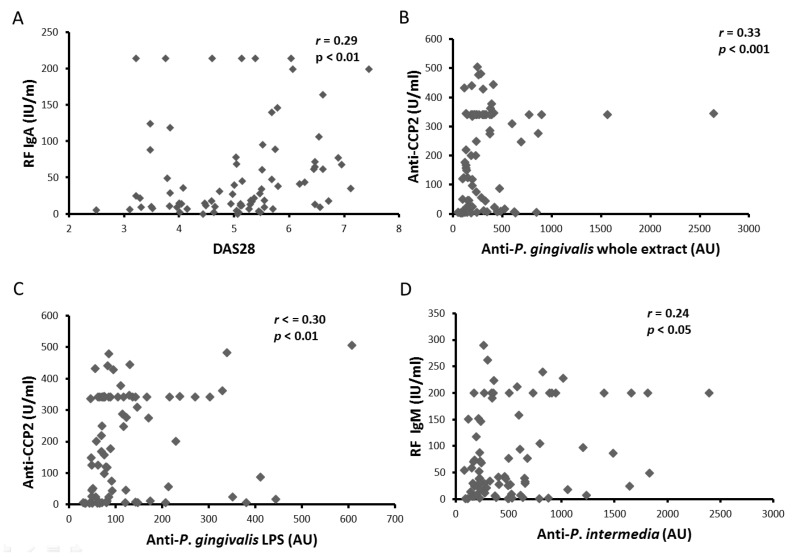
Correlation between clinical and biological parameters at baseline. Correlations between RF IgA and DAS28 (**A**); anti-CCP2 and anti-*P*. *gingivalis* whole extract antibodies (**B**); anti-CCP2 and anti-*P. gingivalis* LPS specific antibodies (**C**); and RF IgG and anti-*P. intermedia* antibodies (**D**). RF: rheumatoid factors; IU: international units; DAS28: disease activity score 28; AU: arbitrary units; anti-CCP2: anti-cyclic citrullinated petides 2nd generation; *r*: Spearman correlation coefficient; LPS: lipopolysaccharide; U: Unit; IU: International unit; NS: non-significant.

**Figure 3 jcm-08-00751-f003:**
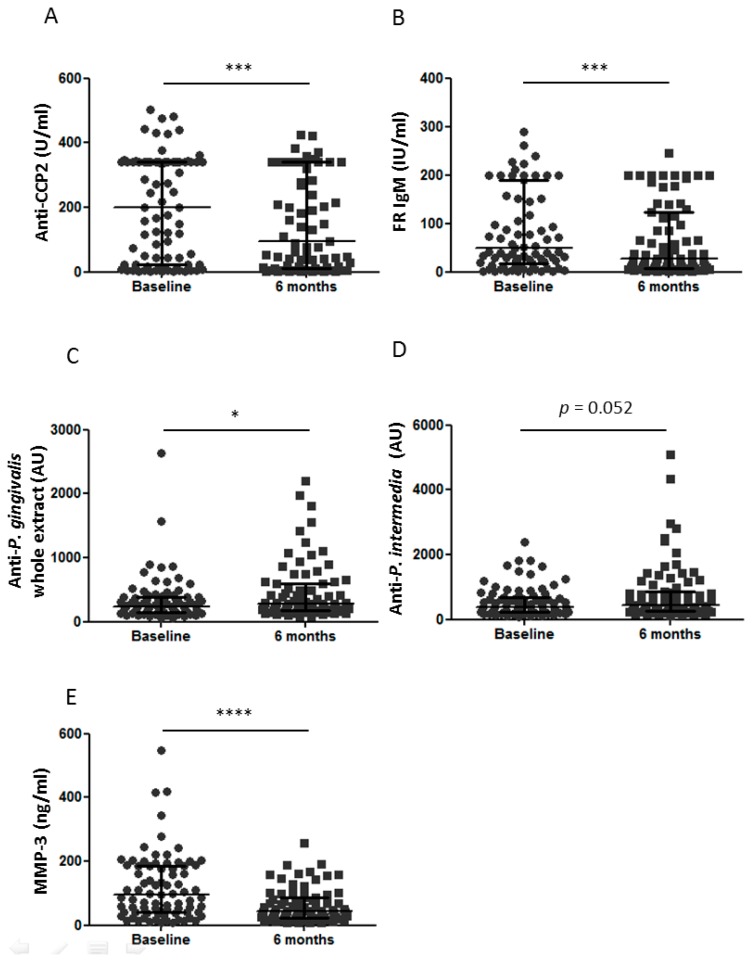
Effect of infliximab treatment on biological markers. Anti-CCP2 (**A**), RF IgM (**B**), anti-*P*. *gingivalis* antibodies (**C**), anti-*P. intermedia* antibodies (**D**), and MMP3 (**E**) were evaluated at baseline and after 6 months of infliximab therapy. Dots represent results for each patient, with values at baseline in round and at 6 months of infliximab therapy in square. From the bottom up, the bars indicate the interquartile range and the median. Anti-CCP2: anti-cyclic citrullinated petides 2nd generation; RF: rheumatoid factor; MMP-3: metalloproteinase 3; AU: arbitrary units; IU: International units; *: *p* < 0.05; ***: *p* < 0.001; ****: *p* < 0.0001.

**Figure 4 jcm-08-00751-f004:**
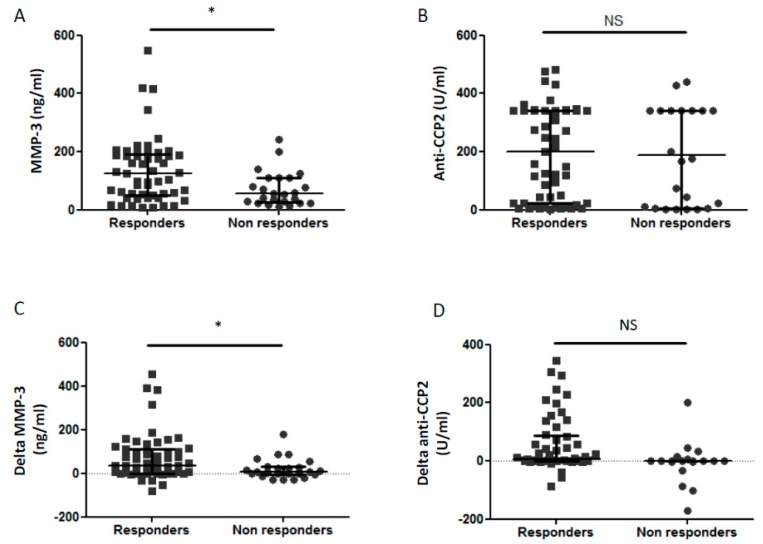
Predictive factors for clinical response to infliximab therapy. Baseline values of MMP-3 (**A**) and anti-CCP2 (**B**) were represented. Delta MMP-3 (**C**) and delta anti-CCP2 (**D**) represent difference of MMP-3 and ACPA (baseline value minus 6 month value) and are shown according to clinical response to infliximab. Dots represent results for each patient, with responders in square (■) and non-responders in round (●). From the bottom up, the bars indicate the interquartile range and the median. Response to infliximab treatment was defined by an improvement > 1.2 of DAS28 at 6 months. MMP-3: matrix metalloproteinase-3; anti-CCP2: anti-cyclic citrullinated petides 2nd generation; *: *p* < 0.05; NS: non significant.

**Table 1 jcm-08-00751-t001:** Characteristics of rheumatoid arthritis (RA) patients at baseline and after infliximab therapy.

	Baseline	6 Months	*p* Values
Sex (female/male)	63/16		
Age, years	52.8 (43.3–59.4)		
Disease duration (years)	9 (5–13)		
No destruction, *n* (%)	12 (15.2%)		
Wrist destruction, *n* (%)	56 (70.9%)		
Periodontal disease, *n* (%)	51 (64.6%)		
DAS28	5.1 (4.1–5.7)	3.5 (2.7–4.3)	<0.0001
ESR (mm/h)	34 (20–49)	18 (10–32)	<0.0001
CRP (mg/L)	18 (7–33)	7 (2–17)	<0.05
Anti–CCP2 (U/mL)	97 (9–275)	43 (5–189)	<0.001
RF IgM (IU/mL)	38 (11–96)	22 (6–66)	<0.0001
RF IgA (IU/mL)	25 (10–61)	17 (8–47)	0.0001
MMP–3 (ng/mL)	90 (40–177)	45 (24–91)	<0.0001
Anti–*P. gingivalis* whole extract (AU)	238 (148–377)	274 (173–557)	<0.01
Anti–*P. gingivalis* LPS (AU)	86 (67–146)	97 (77–152)	NS
Anti–*P. intermedia* whole extract (AU)	390 (131–1558)	436 (246–853)	<0.05

Values are indicated as number of patients (%) or median (1st and 3rd quartiles). DAS28: Disease Activity Score 28; ESR: Erythrocyte sedimentation rate; CRP: C-reactive protein; Anti-CCP2: anti-cyclic citrullinated peptides 2nd generation; RF: rheumatoid factor; MMP-3: Matrix metalloproteinase-3; LPS: lipopolysaccharide; U: unit; IU: international units; AU: arbitrary units, NS: non-significant.

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
