# Peer review of "Infliximab Induced a Dissociated Response of Severe Periodontal Biomarkers in Rheumatoid Arthritis Patients"

_jcm, 2019, doi:10.3390/jcm8050751_

Round 1
Reviewer 1 Report
Review of “Infliximab induced a dissociated response of severe periodontal biomarkers in rheumatoid arthritis patients.”
Major comments:
The paper shows data of levels of P. gingivalis antibodies and MMP-3 in RA patients before and after infliximab therapy. The authors state that serum levels of MMP-3, anti-PG and anti-PI are markers of severity of periodontitis. From literature however it in not shown that serum MMP-3 levels correlate with periodontitis, only with RA. MMP-3 has been evaluated in gingival crevicular fluid or in gingival fibroblasts, and here a relation with periodontitis has been demonstrated (ref 17,42). Both papers (ref 20 ,21) that are mentioned to show results of MMP-3 being a marker for PD only show results for RA. So the rationale for using serum MMP-3 as marker for periodontitis is incorrect. In that aspect also the title of the paper is misleading since there are no periodontal biomarkers that are influenced by infliximab.
In table 1 the percentage periodontal disease is mentioned, however by looking at X-rays periodontal damage is scored and not current periodontal disease, as in bleeding on probing and pocket depth. That might explain the higher levels of PD in this RA cohort. Furthermore, periodontal disease of HC, SLE and IBD patients are not given. Also presence of bacterial pathogens is not shown.
It seems a bit strange that only the positive correlations are shown in figure 2, and it must be noted that these correlations are not convincing, low r-values.
The authors do not state how good response to infliximab was defined.(figure 4)
Since occurrence of P. gingivalis is related to RA and P. Intermedia is not, the latter should be considered as a control pathogen. The relation with IgMRf should be discussed as a coincidence finding or better explained.
The fact that anti-PG rise after infliximab may be interesting and related to increased periodontal inflammation. This is however not sufficiently discussed.
Author Response
Major comments:
The paper shows data of levels of P. gingivalis antibodies and MMP-3 in RA patients before and after infliximab therapy. The authors state that serum levels of MMP-3, anti-PG and anti-PI are markers of severity of periodontitis. From literature however it in not shown that serum MMP-3 levels correlate with periodontitis, only with RA. MMP-3 has been evaluated in gingival crevicular fluid or in gingival fibroblasts, and here a relation with periodontitis has been demonstrated (ref 17,42). Both papers (ref 20 ,21) that are mentioned to show results of MMP-3 being a marker for PD only show results for RA. So the rationale for using serum MMP-3 as marker for periodontitis is incorrect. In that aspect also the title of the paper is misleading since there are no periodontal biomarkers that are influenced by infliximab.
We thank the reviewer for this constructive comment. We apologized to miss the appropriate references. We updated with recent references by adding two recent papers in the MMP-3 paragraph of the introduction showing:
1/ an association between MMP-3 polymorphism and PD [1]
2/ high serum level of MMP-3 in patients with both RA and PD patients [2]
1 da Silva M, de Carvalho A, Alves E, et al. Genetic Factors and the Risk of Periodontitis Development: Findings from a Systematic Review Composed of 13 Studies of Meta-Analysis with 71,531 Participants. 2017;2017:1914073. doi:10.1155/2017/1914073
2 Zhao Y, Jin Y, Ren Y, et al. Expression of matrix metalloproteinase-3 in patients with rheumatoid arthritis and its correlation with chronic periodontitis and rheumatoid arthritis. Zhonghua Kou Qiang Yi Xue Za Zhi 2019;54:164–9.
Accordingly, we still support that serum MMP-3 is a biomarker of PD.
In table 1 the percentage periodontal disease is mentioned, however by looking at X-rays periodontal damage is scored and not current periodontal disease, as in bleeding on probing and pocket depth. That might explain the higher levels of PD in this RA cohort.
We thank the reviewer for this helpful comment. We agree to this comment. However, our RA patients are severe with high level of joint destruction RA patients. This can explain the high rate of PD in this particular population.
Furthermore, periodontal disease of HC, SLE and IBD patients are not given. Also presence of bacterial pathogens is not shown.
We thank the reviewer for this constructive comment. We used these other populations only to assess the biomarkers. Unfortunately, we have no way to assess their periodontal status. However, a PD is not known to be associated with these populations.
It seems a bit strange that only the positive correlations are shown in figure 2, and it must be noted that these correlations are not convincing, low r-values.
We thank the reviewer for this comment on data provided. The figure 2 confirms previous data reported to assess the validity of our population. We agree that the r-values are low. This could be due to small sample size.
The authors do not state how good response to infliximab was defined.(figure 4)
We strongly disagree with this comment. This information is already in the body text on Page 5 Paragraph 1 Line 7: “Clinical response to infliximab was defined by a decrease of DAS28>1.2.” and in the Figure of legend of the Figure 4.
Since occurrence of P. gingivalis is related to RA and P. Intermedia is not, the latter should be considered as a control pathogen. The relation with IgM RF should be discussed as a coincidence finding or better explained.
We thank the reviewer for this comment on association between P. intermedia level and RF. That is correct that P. gingivalis is related to RA and P. intermedia is considered as a control. This association between P. intermedia level and RF title was previously reported and we observed it in the rat AIA model [3].
3 Courbon G, Rinaudo-Gaujous M, Blasco-Baque V, et al. Porphyromonas gingivalis experimentally induces periodontis and an anti-CCP2-associated arthritis in the rat. Annals of the Rheumatic Diseases 2019;78:594–9. doi:10.1136/annrheumdis-2018-213697
The fact that anti-PG rise after infliximab may be interesting and related to increased periodontal inflammation. This is however not sufficiently discussed.
We thank the reviewer for this comment. As suggested, we increased the discussion on this point by adding the followed sentences:
“Despite the growing interest for P. gingivalis pre-existing immunity in RA patients, this was the first observation that serum anti-P. gingivalis antibodies were increasing over time during infliximab therapy or bDAMRDS. It is not surprinsing that bDMARD promotes infections including gingival infection. Our results are in line with previous report with infliximab showing increasing of gingival inflammation during infliximab therapy [28].”

Reviewer 2 Report
The work of Rinaudo-Gaujous M et al analyzed the effect of Infliximab on the expression of biomarkers for RA and PD and shox that one of them, MMP-3, can be a predictor of response to therapy. The work is well done and the results are clearly presented. However there are some issues that need to be addressed.
Major points:
1. The correlation between anti-P. gingivalis LPS specific antibody and anti-P. gingivalis LPS specific antibody is no so strong as to be used as the same measurements. Therefore the rest of correlations with other parameters need to be done with both measurements.
2. In Fig 1D the r and the p values are missing. And as there is an outlier value that make difficult to interpret the figure, the authors must include that values in the graph.
3. The authors found the infliximab treatment reduced the MMP-3 levels, but patients with the higher baseline levels of MMP-3 had the better outcome. How the authors explain these contradictory results?
4. In the conclusion section the authors state that “anti-P. intermedia antibody concentrations correlated only with RF concentrations suggesting different immunologic response induced by both oral bacteria”. This conclusion is an overestimation of the correlations performed, then the authors should delete the sentence or providing experiments that support the hypothesis.
Minor points:
1. The authors should improve the introduction about RA, because is not only “characterized by joint and subchondral bone destruction”.
2. In some correlation graphs the p values are above the r values and in other is the other way around. I recommend use the same order in all the graphs.
3. The authors should write the full name of bDMARDS.
Author Response
The work of Rinaudo-Gaujous M et al analyzed the effect of Infliximab on the expression of biomarkers for RA and PD and shox that one of them, MMP-3, can be a predictor of response to therapy. The work is well done and the results are clearly presented. However there are some issues that need to be addressed.
Major points:
1. The correlation between anti-P. gingivalis LPS specific antibody and anti-P. gingivalis LPS specific antibody is no so strong as to be used as the same measurements. Therefore the rest of correlations with other parameters need to be done with both measurements.
We thank the reviewer for this constructive comment. However, we are confusing by this comment. We expected that you mean anti-P. gingivalis LPS specific antibody and anti-P. gingivalis extract specific antibody. We agree that they are not so good for diagnosis purpose. We performed this correlation to test variability of these two assays previously reported in the litterature. Except the Fig 3C, all other data provided both assessments (anti-P. gingivalis LPS specific antibody and anti-P. gingivalis extract specific antibody.
2. In Fig 1D the r and the p values are missing. And as there is an outlier value that make difficult to interpret the figure, the authors must include that values in the graph.
We thank the reviewer for this comment. We are sorry for the confusion. We provided now the R and P values, but we did not expect an outlier. This patient has mainly an antibody against E coli, but not against P. gingivalis.
3. The authors found the infliximab treatment reduced the MMP-3 levels, but patients with the higher baseline levels of MMP-3 had the better outcome. How the authors explain these contradictory results?
We thank the reviewer for this comment. The easy answer is to consider that MMP3 is biomarker of RA disease activity. So we add the following sentence in the discussion “Further studies are required before to consider MMP-3 as a validate biomarker to predict clinical response usable in the daily practice”.
4. In the conclusion section the authors state that “anti-P. intermedia antibody concentrations correlated only with RF concentrations suggesting different immunologic response induced by both oral bacteria”. This conclusion is an overestimation of the correlations performed, then the authors should delete the sentence or providing experiments that support the hypothesis.
We thank the reviewer for this comment. The correlation between anti-P. intermedia antibody concentrations correlated only with RF concentrations was previously reported in a rat model [1].
Minor points:
1. The authors should improve the introduction about RA, because is not only “characterized by joint and subchondral bone destruction”.
As suggested by the reviewer, we added more data on the RA with the updated following sentence: “Rheumatoid arthritis (RA) is the most frequently chronic inflammatory joint disease characterized by synovial hypertrophy and inflammation with joint and subchondral bone destruction, which correlates with disability and loss of function [1].”
2. In some correlation graphs the p values are above the r values and in other is the other way around. I recommend use the same order in all the graphs.
As suggested by the reviewer, we checked the same presentation for all figures.
3. The authors should write the full name of bDMARDS.
As suggested by the reviewer, we defined bDMARDs as biologic disease modifying anti-rheumatic drugs on Page 4.

Reviewer 3 Report
The paper "Infliximab induced a dissociated response of severe periodontal biomarkers in rheumatoid arthritis patients" is well written and well designed. The authors indicate a significant relationship between periodontitis (and related markers) and the development of RA and the response to infliximab.In this context, the study may be useful in everyday clinical practice.
Minor comments:
In the description of Table 1. the LPS abbreviation should be expanded
Author Response
The paper "Infliximab induced a dissociated response of severe periodontal biomarkers in rheumatoid arthritis patients" is well written and well designed. The authors indicate a significant relationship between periodontitis (and related markers) and the development of RA and the response to infliximab.In this context, the study may be useful in everyday clinical practice.
We thank the reviewer for its nice comment.
Minor comments:
In the description of Table 1. the LPS abbreviation should be expanded
We thank the reviewer for this comment. We expanded the LPS as asked.
Round 2
Reviewer 1 Report
The comments have been sufficiently addressed, although there still are questions left about presentation of the data. However, the paper has significantly improved.
Author Response
We thank the reviewer for this positive comment on the revised manuscript. We are wondering about the remark on “presentation of the data”. We expected that this remark referred to the figure 2. However, we previously address this point on the previous revised manuscript.
Reviewer 2 Report
Although some questions were answered, there are another ones that still need to be improved.
Figure 1D. This figure is more confusing now, as the authors claim that "No crossreaction was observed in our assay since high anti-P. gingivalis antibody concentrations remained negative for anti-E. coli antibody". However there is a positive correlation between the presence of both antibodies (at least as strong as in Fig 1B and C). Then this correlation suggest that there is a cross-reaction in the assay.
Page 8, line 284. The sentence "Thus high antibody concentrations against oral pathogens were specific for RA disease" is incorrect. SLE patients have higher specific antibodies compared to IBD patients, Therefore statistical compilations must be done and the sentence must be changed.
Page 11, line 380: Taken all together, our data support the specific implication of P. gingivalis in established RA pathogenesis with induction of anti-CCP2 in response to gingival citrullinated proteins. This sentence is an overestimation of the results, as is based in the weak correlations between the anti-CCP2 antibody levels and the anti-P. gingivalis whole extract (r=0,33) and LPS specific (r=0,30) antibody levels and the citrullination ability of P. gingivalis.
Moreover, the fact that infliximad treatment reduced the anti-CCP2 levels but induced the levels of anti-P. gingivalis suggest that the presence of anti-CCP2 antibodies is not dependent of the P. gingivalis-induced citrullination. Therefore the authors should demonstrate their hypothesis or remove the sentence.
Author Response
Although some questions were answered, there are another ones that still need to be improved.
We thank the reviewer for its positive comment on the revised work performed.
Figure 1D. This figure is more confusing now, as the authors claim that "No crossreaction was observed in our assay since high anti-P. gingivalis antibody concentrations remained negative for anti-E. coli antibody". However there is a positive correlation between the presence of both antibodies (at least as strong as in Fig 1B and C). Then this correlation suggest that there is a cross-reaction in the assay.
We thank the reviewer for its constructive comment and we tried to clarify this big issue. The figure 1D was provided to prevent criticism about the cross reactivity with all digestive pathogens. In the Figure 1D, we provided data on the correlation between levels of anti-LPS E. coli and anti-LPS P. gingivalis antibodies. The graph shows clearly in one patient a strong immunity for E coli without immunity against P. gingivalis. Some patients had huge immunity against P. gingivalis without immunity against E. coli. Accordingly, we still believe that immunity against oral germs (P. gingivalis and P. intermedia) did not crossreact with immunity against E. coli (more specific of bowel). Furthermore, the structure of LPS of P. gingivalis is largely different to other LPS.
In the Fig 1B, we described the positive correlation between LPS and bacteria extract for P. gingivalis to compare these two ways to assess immunity against P. gingivalis, as expected.
In the Fig 1C, we described the positive correlation between two oral germs: P. gingivalis and P. intermedia, as expected.
Accordingly, we updated mainly the text in the legends of Figure. After reading of the manuscript, we think the text is enough clear in the manuscript.
Page 8, line 284. The sentence "Thus high antibody concentrations against oral pathogens were specific for RA disease" is incorrect. SLE patients have higher specific antibodies compared to IBD patients, Therefore statistical compilations must be done and the sentence must be changed.
We thank the reviewer for its comment. We agreed with the reviewer that data presented are quite strange. We checked the raw data and performed new statistical analysis as suggested. We observed high level for both antibodies (anti-P. gingivalis and anti-P. intermedia) in RA and SLE patients compared to IBD. Accordingly, we updated the manuscript by “Anti-P. gingivalis and anti-P. intermedia antibody concentrations were lower in IBD and SLE patients and established RA patients (Fig. 1E&F; P<0.001 for P. gingivalis and P. intermedia in RA and SLE vs. IBD).Thus high antibody concentrations against oral pathogens were specific for RA or SLE diseases involving joint.”
Page 11, line 380: Taken all together, our data support the specific implication of P. gingivalis in established RA pathogenesis with induction of anti-CCP2 in response to gingival citrullinated proteins. This sentence is an overestimation of the results, as is based in the weak correlations between the anti-CCP2 antibody levels and the anti-P. gingivalis whole extract (r=0,33) and LPS specific (r=0,30) antibody levels and the citrullination ability of P. gingivalis.
We thank the reviewer for its comment. Since our analysed population was an established RA, we conclude on this population. We agree that the correlation in quite weak, but still significant in a quite small population of established RA. However, this statement is partially based on our results integrated with our previous data on a rodent model (ref 5), where we clearly demonstrated the specificity of P. gingivalis to induce anti-CPP2 and then arthritis. To clarify our statement, we added on the revised version “compared to P. intermedia” in the discussed sentence.
Moreover, the fact that infliximab treatment reduced the anti-CCP2 levels but induced the levels of anti-P. gingivalis suggest that the presence of anti-CCP2 antibodies is not dependent of the P. gingivalis-induced citrullination. Therefore the authors should demonstrate their hypothesis or remove the sentence.
We thank the reviewer for its comment, but disagree for the interpretation. Previous data on non-surgical therapy of periodontal disease observed the opposite effect with a decreased of anti-P. gingivalis levels, but without effect on anti-CCP2 levels (ref 32). Since previous report observed an increased gingival inflammation induced by infliximab (ref), we expected an increasing of anti-P. gingivalis and anti- P. intermedia levels. So it was not surprising to observe this dissociation since this was induced by infliximab. As discussed previously, the fact is that recent data suggest that P. gingivalis-induced citrullination in early RA or before arthritis onset. After this phase, the two processes can diverge with still some interaction.
We added the dissociated effect on the introduction section and discussed this point as follow: “This kind of dissociation response was also observed after non-surgical therapy for periodontal disease with a decreased of anti-P. gingivalis levels without effect on anti-CCP2 levels [32]. This dissociated response to infliximab of anti-P. gingivalis and anti-CCP2 concentrations reinforced role of P. gingivalis only in anti-CCP2 induction.”